# Redefining part-of-speech classes with distributional semantic models

## Abstract

This paper studies how word embeddings trained on the British National Corpus interact with part of speech boundaries. We experiment with training classifiers for predicting PoS tags for words based on their embeddings. The results show that the distributional vectors do contain information about PoS affiliation. This approach allowed us to discover word groups with distributional patterns different from other words of the same part of speech. This often reveals hidden inconsistencies of the annotation process or guidelines. At the same time, it supports the notion of 'soft' or 'graded' part of speech affiliations. Finally, we found out that information about PoS is distributed among dozens of vector components, not limited to only one or two features.

## 1 Introduction

Parts of speech (PoS) are useful abstractions, but still abstractions. Boundaries between them in natural languages are flexible. Sometimes, large open classes of words are situated on the verge between several parts of speech: for example, participles in English are in many respects both verbs and adjectives. In other cases, closed word classes 'intersect' between themselves: it is often difficult to tell a determiner from a possessive pronoun, etc. As (Houston, 1985) puts it, '*Grammatical categories exist along a continuum which does not exhibit sharp boundaries between the categories*'.

When annotating natural texts for parts of speech, the choice of a PoS tag in many ways depends on the human annotators themselves, but also on the quality of linguistic conventions behind the division into different word classes. That is why there were always attempts to refine the definitions of parts of speech and to make them more 'real' and data-driven, produced from corpora of real texts: see, among others, the seminal work of (Biber et al., 1999). The aim of such attempts is to identify clusters of words occurring naturally and corresponding to what we usually call 'parts of speech'. One of the main distance metrics that can be used in detecting such clusters is a distance between distributional features of words (their contexts in a reference training corpus).

In this paper, we test this approach using predictive models developed in the field of distributional semantics. Recent achievements in training distributional models of language using machine learning allows for robust representations of natural language semantics created in a completely unsupervised way, using only large corpora of raw text. Relations between dense word vectors (embeddings) in the resulting vector space are of course mostly semantic. But can they be used to discover something new about grammar and syntax, particularly parts of speech? Do learned semantic vectors help here? Below we show that such models do contain a lot of interesting data related to PoS classes.

The rest of the paper is organized as follows. In Section 2 we briefly cover the previous work on the subject of parts of speech and distributional models. Section 3 describes data processing and the training of a PoS predictor based on word embeddings. In Section 4 errors of this predictor are analyzed and insights gained from them described. Section 5 introduces an attempt to build a full-fledged PoS tagger within the same approach. It also analyzes the correspondence between particular vector components and PoS affiliation, before we conclude in Section 6.

## 2 Related work

Traditionally three types of criteria are used to distinguish different parts of speech: formal (or morphological), syntactic (or distributional) and semantic (Aarts and McMahon, 2008). Arguably, syntactic and semantic criteria are not much different from each other, if one follows the famous distributional hypothesis stating that meaning is determined by context (Firth, 1957). Below we show that unsupervised distributional semantic models obviously contain data related to parts of speech.

For several years already it has been known that some information about morphological word classes is indeed stored in distributional models; (Tsuboi, 2014) even employed it to improve PoS-tagging. Words belonging to different parts of speech possess different contexts: in English, articles are typically followed by nouns, verbs are typically accompanied by adverbs and so on. It means that during the training stage, words of one PoS should theoretically cluster together or at least their embeddings should retain some similarity allowing to separate them from words belonging to other parts of speech.

(Mikolov et al., 2013b) showed that there are indeed also regular relations between words from different classes: the vector of '*Brazil*' is related to '*Brazilian*' in the same way as '*England*' is related to '*English*' and so on. Later, (Liu et al., 2016) demonstrated how words of the same part of speech cluster into distinct groups in a distributional model, and (Tsvetkov et al., 2015) proved that dimensions of distributional models are correlated with different linguistic features, releasing the evaluation dataset based on this.

It seems that one can infer data about PoS classes of words from embedding models. But then, it can be useful for deeper analysis of part of speech boundaries, leading to discovery of separate words or whole classes that tend to behave 'strangely'. Discovering such cases is one possible way to improve performance of existing automatic PoS taggers (Manning, 2011). These 'outliers' may signal the necessity to revise the annotation strategy or classification system in general. Section 3 describes the process of constructing typical PoS clusters and detecting words which seem to belong to a cluster different from their traditional annotation.

## 3 Part of speech clusters in distributional models

Our hypothesis is that for the majority of words their part of speech can be inferred from their embeddings in a distributional model. This inference can be considered a classification problem: we are to train an algorithm that takes a word vector as input and outputs its part of speech. If the word embeddings do contain PoS-related data, the properly trained classifier will correctly predict PoS tags for the majority of words: it means that these lexical entities conform to a dominant distributional pattern of their part of speech class. At the same time, the words for which the classifier outputs *incorrect* predictions, are expected to be 'outliers', with different distributional patterns, different from other words in the same class. These cases are the points of linguistic interest, and in the rest of the paper we mostly concentrate on them.

To test the initial hypothesis, we used the XML Edition of British National Corpus (BNC), a balanced and representative corpus of English language of about 98 million word tokens in size. As stated in the corpus documentation, '*it was* [PoS-]*tagged automatically, using the CLAWS4 automatic tagger developed by Roger Garside at Lancaster, and a second program, known as Template Tagger, developed by Mike Pacey and Steve Fligelstone*' (Burnard, 2007). The corpus authors report a precision of 0.96 and recall of 0.99 for their tools, based on a manually checked sample. For this research, it is important that BNC is an established and well-studied corpus of English with PoS-tags and lemmas assigned to all words.

We produced a version of BNC where all the words were replaced with their lemmas and PoS-tags converted into the Universal Part-of-Speech Tagset (Petrov et al., 2012)[1]. Thus, each token was represented as a concatenation of its lemma and PoS tag (for example, '*love_VERB*' and '*love_NOUN*' yield different word types). The mappings between BNC tags and Universal tags were created manually by us and released online[2]. We worked with the following 16 Universal tags: **ADJ, ADP, ADV, AUX, CONJ, DET, INTJ, NOUN, NUM, PART, PRON, PROPN, SCONJ, SYM, VERB, X** (tokens marked with PUNCT tag were excluded).

---

[1]We used the latest version of the tagset available at http://universaldependencies.org
[2]Anonymized

Then, a *Continuous Skipgram* embedding model (Mikolov et al., 2013a) was trained on this corpus, using a vector size of 300, 10 negative samples, a symmetric window of 2 words, no down-sampling, and 5 iterations over the training data. Words with corpus frequency less than 5 were ignored. This model was then taken to represent the semantics of the words it contained. But at the same time, for each word, a gold standard PoS tag is known (from the BNC annotation). It means that is is possible to test how good the word embeddings are in grouping words according to their parts of speech.

To this end, we extracted vectors for the 10 000 most frequent words from the resulting model (roughly, these are the words with corpus frequency more than 500). Then, these vectors were used to train a simple logistic regression multinomial classifier aimed to predict the word's part of speech[3].

Note that during training (and subsequent testing), each word's vector was used several times, proportional to frequency of the word in the corpus, so the classifier was trained on 177 343 (sometimes repeating) instances, instead of the original 10 000. This was done to alleviate the classifier's part of speech bias. There are much fewer word types in the closed PoS classes (pronouns, conjunctions, etc.) than in the open ones (nouns, verbs, etc.), so without considering word frequency, the model does not have a chance to learn good predictors for 'rare' classes and ends up never predicting them. At the same time, words from closed classes occur very frequently in the running text, so after 'weighting' training instances by corpus frequency, the balance is restored and the classifier model has enough training instances to learn to predict closed PoS classes as well. As an additional benefit, by this modification we make frequent words from all classes to be more 'influential' in training the classifier.

The resulting classifier showed a weighted average F-score equal to 0.979 with 10-fold cross-validation on the training set. This is a significant improvement over the *majority class* baseline classifier (classify everything as nouns), which showed an F-score of 0.07 and over the *one-feature* baseline classifier (classify using only one

---

[3]It is important that we applied classification, not clustering here. Attempts to naively cluster word vectors into the number of clusters equal to the number of PoS tags inescapably failed.

vector dimension with maximum F-value in relation to class tags), with F-score equal to only 0.22. Thus, the results support the hypothesis that word embeddings contain information allowing to group words together based on their parts of speech. At the same time, we see that this information is not restricted to some particular vector component: rather, it is distributed among several axis of the vector space.

After training the classifier, we were able to use it to detect 'outlying' words in the BNC (judging by the distributional model). So as not to experiment on the same data we had trained our classifier on, we compiled another test set of 17 000 vectors for words with BNC frequencies between 100 and 500. They were weighted by word frequencies in the same way as the training set, and the resulting test set contained 30 710 instances. Then, we predicted parts of speech for these words using our classifier and evaluated its performance on them. The results on totally unseen data were not much worse, with an F-score equal to 0.91.

Furthermore, to make sure that the results can potentially be extended to other texts, we applied the trained classifier to all lemmas from the human-annotated Universal Dependencies English Treebank (Silveira et al., 2014). The words not present in the distributional model were omitted (they sum to 27% of word types and 10% of word tokens). The classifier showed an F-Score equal to 0.99, further demonstrating the robustness of the classifier.

In sum, the vast majority of words are classified correctly, which means that their embeddings enables detecting their parts of speech. In fact, one can even visualize 'prototypical' vectors for each PoS by simply averaging vectors of words belonging to this part of speech. We did this for 10 000 words from our training set.

Plots for prototypical vectors of coordinating and subordinating conjunctions are shown in the Figures 1 and 2 respectively. Even visually one can notice a very strongly expressed feature near the '100' mark in the horizontal axis. In fact, this is vector component number 94, and it is indeed an idiosyncratic feature of conjunctions: none of other parts of speech shows such a property. More details about what vector components are relevant to part of speech affiliation are given in Section 5.

With prototypical PoS vectors we can even find out how similar different parts of speech are to

**Figure 1.** Prototypical embedding for coordinating conjunctions

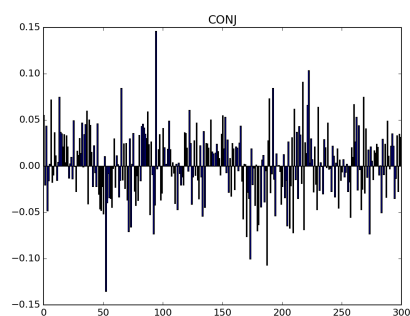

**Figure 2.** Prototypical embedding for subordinating conjunctions

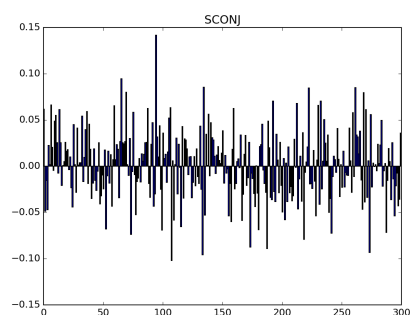

each other, by simply measuring cosine similarity between them. If we rank PoS pairs according to their similarity, what we see is that nominative parts of speech are close to each other, determiners and pronouns are also similar, as well as prepositions and subordinating conjunctions, everything quite in accordance with language intuition. It is interesting that proper nouns are not much similar to common nouns, with cosine similarity between them only 0.67 (even adverbs are closer). As we show below, this helps the model to successfully separate the former from the latter.

Despite generally good performance of the classifier, if we look at our BNC test set, 1741 word types (about 10% of the whole test set vocabulary) were still classified incorrectly. Thus, they are somehow dissimilar to 'prototypical' words of their parts of speech. These are the 'outliers' we were after. We investigate the patterns found among them in the next section.

## 4   Not from this crowd: analyzing outliers

We filtered out mis-classified word types with 'X' gold annotation (they are mostly foreign or nonsense words). This left us with 1558 words; the classifier assigned them part of speech tags

**Table 1.** Most frequent PoS mis-classifications of the distributional predictor

| Amount (word types) | Actual PoS | Predicted PoS |
|---|---|---|
| 347 | PROPN | NOUN |
| 313 | ADJ | NOUN |
| 190 | NOUN | ADJ |
| 91 | NOUN | PROPN |
| 87 | PROPN | ADJ |
| 57 | VERB | ADJ |
| 55 | NOUN | NUM |
| 52 | NUM | NOUN |
| 45 | NUM | PROPN |
| 28 | ADV | PROPN |
| 25 | ADV | NOUN |
| 25 | ADJ | PROPN |
| 20 | ADV | ADJ |

different from the ones in the BNC. It probably means that these words' distributional patterns differ somehow from the 'mainstream', and they tend to exhibit behavior similar to another part of speech. Table 1 shows the most frequent mis-classification cases, together accounting for more than 85% of errors.

Additionally, we ranked mis-classification cases by 'part of speech coverage', that is by the ratio of the words belonging to a particular PoS for which our classifier outputs this particular type of mis-classification. For example, proper nouns mis-classified as common nouns constitute the most numerous error type in Table 1, but in fact only 9% of all proper nouns in the test set were mis-classified in this way. There are parts of speech with a much larger portion of word-types predicted erroneously: e.g., 22% of subordinate conjunctions were classified as adverbs. Table 2 lists error types with the highest coverage (we excluded error types with absolute frequency equal to 1, as it is impossible to speculate on solitary cases).

We now describe some of the interesting cases. Almost 30% of error types (judging by absolute amount of mis-classified words) consist of proper nouns predicted to be common ones and vice versa. These cases do not tell us anything new, as it is obvious that distributionally these two classes of words are very similar, take the same syntactic contexts and hardly can be considered differ-

ent parts of speech at all. At the same time, it is interesting that the majority of proper nouns in the test set (88%) was correctly predicted as such. It means that in spite of contextual similarity, the distributional model has managed to extract features typical for proper names. Errors mostly cover comparatively rare names, such as '*luftwaffe*', '*stasi*', '*stonehenge*', or '*himalayas*'. Our guess is that the model was just not presented with enough contexts for these words to learn meaningful representations. Also, they are mostly not personal names but toponyms or organization names. Most probably, the model has trained to distinguish proper names by contexts like '*My name is*', etc, obviously not appropriate here.

Another 30% of errors are due to vague boundaries between nominal and adjectival distribution patterns in English: nouns can be modified by both (it seems that cases where a proper noun is mistaken for an adjective are often caused by the same factor). Words like '*materialist_NOUN*', '*starboard_NOUN*' or '*hypertext_NOUN*' are tagged as nouns in the BNC, but they often modify other nouns, and their contexts are so 'adjectival' that the distributional model actually assigned them semantic features highly similar to those of adjectives. Vice versa, '*white-collar_ADJ*' (an adjective in BNC) is surely a noun from the point of view of our model. Indeed, there can be contradicting views on the correct part of speech for this word in phrases like '*and all the other white-collar workers*'. Thus, in this case the distributional model highlights the already known similarity between two word classes.

The cases with verbs mistaken for adjectives seem to be caused mostly by passive participles ('*was overgrown*', '*is indented*', ''), which intuitively are indeed very adjective-like. So, this gives us a set of verbs dominantly (or almost exclusively, like '*to intertwine*' or '*to disillusion*') used in passive. Of course, we will hardly announce such verbs to be adjectives based on that evidence, but at least we can be sure that this subclass of verbs is clearly semantically and distributionally different from other verbs.

The next numerous type of errors consists of common nouns predicted to be numerals. A quick glance at the data reveals that 90% of these 'nouns' are in fact currency amounts and percentages ('*£70*', '*33%*', '*$1*', etc). It seems pretty logical to classify these as numerals, despite of them containing some kind of nominative entities inside. Judging by the classifier's decisions, their contexts do not differ much from those of simple numbers, and their semantics is similar. The Universal Dependencies Treebank is more consistent in this respect: it separates entities like '*1$*' into two tokens: a numeral (NUM) and a symbol (SYM). Consequently, when our classifier was tested on words from the UD Treebank, there was only one occurrence of this type of error.

Related to this is the inverse case of numerals predicted to be common or proper nouns. It is interesting that this error type is also quite massive in its coverage: if we combine numerals predicted to be common and proper nouns, we will see that 17% of all numerals in the test set were subject to this error. The majority of these 'numerals' are years ('*1804*', '*1776*', '*1822*') and decades ('*1820s*', '*60s*' and even '*twelfths*'). Intuitively, such entities do indeed functions as nouns ('*I'd like to return to the sixties*'). Anyway, it is difficult to invent a persuasive reason for why '*fifty pounds*' should be tagged as a noun, but '*the year 1776*' as a numeral. So, this points to possible (minor) inconsistencies in the annotation strategy of the BNC. Note that a similar problem exists in the Penn Treebank as well (Manning, 2011).

Adverbs classified as nouns (53 words in total for both common and proper nouns) are possibly the ones often followed by verbs or appearing in company of adjectives (examples are '*ultra*' and '*kinda*'). This made the model treat them as close to the nominative classes. Interestingly, most 'adverbs' predicted to be proper nouns are time indicators ('*7pm*', '*11am*'); this also raises questions about what adverbial features are present in these entities. Once again, the UD Treebank does not tag them as adverbs.

The cases we described above revealed some inconsistencies in the BNC annotation. However, it seems that with adverbs mistaken for adjectives, we actually found a systematic error in the BNC tagging: these cases are mostly connected to adjectives like '*plain*', '*clear*' or '*sharp*' (including comparative and superlative forms) erroneously tagged in the corpus as adverbs. These cases are not rare: just the three adjectives we mentioned alone appear in the BNC about 600 times with an adverb tag, mostly in phrases of the kind '*the author makes it plain that...*'. Sometimes these to-

**Table 2.** Coverage of mis-classifications (from all word types of this PoS) with distributional predictor

| Coverage | Actual PoS | Predicted PoS | Absolute amount |
|---|---|---|---|
| 0.22 | SCONJ | ADV | 2 |
| 0.17 | INTJ | PROPN | 8 |
| 0.11 | ADP | ADJ | 3 |
| 0.09 | ADJ | NOUN | 313 |
| 0.09 | PROPN | NOUN | 347 |
| 0.09 | NUM | NOUN | 52 |
| 0.08 | NUM | PROPN | 45 |

kens are tagged as ambiguous, and the adjective tag is there as a second variant; however, the corpus documentation states that in such cases the first variant is always more likely. Thus, distributional models can actually detect outright errors in PoS-tagged corpora, when incorrectly tagged words strongly tend to cluster with another part of speech. In the UD treebank such examples can also be observed, but they are much fewer and more 'adverbial', like '*it goes **clear** through*'.

Some of the entries from Table 2 were already covered above, except the first three cases. They are related to closed word classes (functional words), that's why the absolute number of influenced word types is low, but the coverage (ratio of all words of this PoS) is quite high.

First, of 9 distinct subordinate conjunctions in the test set, two were predicted to be adverbs. This is not surprising, as these words are '*seeing*' and '*immediately*'. For '*seeing*' the prediction seems to be just a random guess (the prediction confidence was as low as 0.3), but with '*immediately*' the classifier was actually more correct than the BNC tagger (the prediction confidence was about 0.5). In BNC, these words are mostly tagged as subordinate conjunctions in cases when they are in the beginning of sentences ('*Immediately, she lowered the gun*'). The other words marked as SCONJ in the test set are really such, and the classifier made correct predictions matching the BNC tags.

Interjections mistaken for proper names do not seem very interpretable (examples are '*gee*', '*oy*' and '*farewell*'). At the same time, 3 prepositions predicted to be adjectives clearly form a separate group: they are '*cross*', '*pre*' and '*pro*'. They are

not often used as separate words, but when they are ('*Did anyone encounter any trouble from Hibs fans in Edinburgh **pre** season?*'), they are very close to adjectives or adverbs, so the predictions of the distributional classifier once again suggest shifting parts of speech boundaries a bit.

Error analysis on the vocabulary from the Universal Dependencies Treebank showed pretty much the same results, except for some differences mentioned above.

There exists another way to retrieve this kind of data: to process gold standard data with a mainstream PoS tagger and analyze the resulting confusion matrix. We tested this approach by processing the whole BNC with the Stanford PoS Tagger (Toutanova et al., 2003). Note that as an input to the tagger we used not the whole sentences from the corpora, but separate tokens, to mimic our workflow with the distributional predictor. Prior to this, BNC tags were converted to the Penn Treebank tagset[4] to match the output of the tagger. As we are interested in coarse, 'overarching' word classes, inflectional forms were merged into one tag (for example plural and singular nouns NNS and NN were considered to belong to one noun class NN, etc). That was easy to accomplish by dropping all characters of the tags after the first two (excluding proper noun tags, which were all converted to NNP).

Analysis of the confusion matrix (cases where the tag predicted by the Stanford tagger was different from the BNC tag) revealed the most frequent error types shown in Table 3. Despite similar top positions of errors types 'proper noun predicted as common noun' and 'nouns and adjectives mistaken for each other', there are also very frequent errors of types 'verb to noun' and 'adjective to verb', not observed in the distributional confusion matrix (Table 1). We would not be able to draw the same insights that we drew from the distributional confusion matrix: the case with verbs mistaken for adjective is ranked only 12th, adverbs mistaken for nouns - 13th, etc.

Table 4 shows top mis-classification types by their word type coverage. Once again, interesting cases we discovered with the distributional confusion matrix (like subordinating conjunctions mistaken for adverbs and prepositions mistaken for adjectives) did not show up. Obviously, a lot of other insights can be extracted from the Stanford

---

[4] https://www.cis.upenn.edu/~treebank/

**Table 3.** Most frequent PoS mis-classifications using the Stanford tagger

| Amount (word types) | Actual PoS | Predicted PoS |
|---|---|---|
| 172675 | NNP | NN |
| 47202 | VB | NN |
| 40218 | JJ | NN |
| 24075 | NN | JJ |
| 9723 | JJ | VB |

**Table 4.** Coverage of mis-classifications (from all word types of this PoS) with the Stanford tagger

| Coverage | Actual PoS | Predicted PoS | Absolute amount |
|---|---|---|---|
| 0.91 | NNP | NN | 172675 |
| 0.8 | UH | NN | 576 |
| 0.79 | DT | NN | 217 |
| 0.78 | EX | JJ | 11 |
| 0.78 | PR | NN | 517 |

Tagger errors (and it was studied before), but it seems that employing a distributional predictor reveals different error cases and thus might be a useful tool.

To sum it up, analysis of 'boundary cases' detected by a classifier trained on distributional vectors, indeed reveals sub-classes of words lying on the verge between different parts of speech. It also allows for quickly discovering systematic errors or inconsistencies in PoS tags, whether they be automatic or manual. Thus, discussions about PoS boundaries definitely should take into consideration this data (expanded and revised).

## 5 Distributional vectors as part-of-speech predictors

In the experiment described in the previous section, we used the model trained on words concatenated with their PoS tags. Thus, our 'classifier' was a bit artificial in that it demanded a word plus a tag as an input, and then its output is a judgment about what part of speech is most applicable to this combination from the point of view of the BNC distributional patterns. This was not a problem for us, as our aim was exactly to discover lexical outliers.

But is it possible to construct a proper predictor in the same way, which is able to predict a PoS tag for a word without any pre-existing tags as hints? Preliminary experiments seem to indicate that it is.

We trained a *Continuous Skipgram* distributional model on the BNC lemmas without PoS tags. After that, we constructed a vocabulary of all unambiguous lemmas from the UD Treebank training set. 'Unambiguous' here means that the lemma either was always tagged with one and the same PoS tag in the Treebank, or has one 'dominant' tag, with frequencies of other PoS assignments not exceeding 1/2 of the dominant assignment frequency. Our hypothesis was that these words are prototypical examples of their PoS classes, with corresponding prototypical features most pronounced. We also removed words with frequency less than 10 in the Treebank. This left us with 1564 words from all Universal Tag classes (excluding PUNCT, X and SYM, as we hardly want to predict punctuation or symbol tag).

Then the same simple logistic regression classifier was trained on the distributional vectors from the model for these 1564 words only, using UD Treebank tags as class labels (the training instances were again weighted proportionally to the words' frequencies in the Treebank). The resulting classifier showed an accuracy of 0.938 after 10-fold cross-validation on the training set.

We then evaluated the classifier on tokens from the UD Treebank test set. Now the input to the classifier consisted of lemma only. Lemmas which were missing from the model's vocabulary were omitted (860 of a total of 21759 tokens in the test set). It reached an accuracy of 0.84 (weighted precision 0.85, weighted recall 0.84).

These numbers may not seem very impressive in comparison with the performance of modern state-of-the-art PoS taggers. However, one should remember that this classifier knows absolutely nothing about a word's context in the current sentence. It assigns PoS tags based solely on the proximity of the word's distributional vector in an unsupervised model to those of prototypical PoS examples. The classifier was in fact based only on knowledge of what words occurred in the BNC near other words within a symmetric window of 2 word to the left and to the right. It did not even have access to the information about exact word order within this sliding window, which makes its performance even more impressive.

**Figure 3.** Classifier accuracy depending on the number of used vector components ($k$)

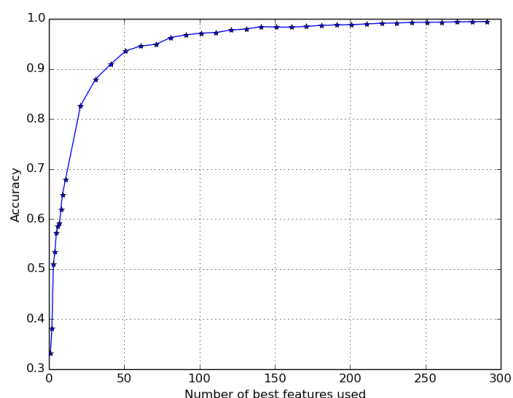

It is also interesting that one needs as few as a thousand example words to train a decent classifier. Thus, it seems that PoS affiliation is expressed quite strongly and robustly in word embeddings. It can be employed, for example, in preliminary tagging of large corpora of resource-poor languages. Only a handful of non-ambiguous words need be manually PoS-tagged, and the rest is done by a distributional model trained on the corpus.

To find out how many features are important for the classifier, we used the same training and test set, and ranked all embedding components (features, vector dimensions) by their ANOVA F-value related to PoS class. Then we successively trained the classifier on increasing amounts of top-ranked features (top $k$ best) and measured the training set accuracy.

The results are shown in Figure 3. One can see that the accuracy smoothly grows with the number of used features, eventually reaching almost ideal performance on the training set. It is difficult to define the point where the influence of adding features reaches a plateau; it may lie somewhere near $k = 100$. It means that the knowledge about PoS affiliation is distributed among at least one hundred components of the word embeddings, quite consistent with the underlying idea of embedding models.

One might argue that the largest gap in performance is between $k = 2$ and $k = 3$ (from 0.38 to 0.51) and thus most PoS-related information is contained in the 3 components with the largest F-value (in our case, these 3 features were components 31, 51 and 11). But an accuracy of 0.51 is certainly not an adequate result, so even if im-portant, these components are not sufficient to robustly predict part of speech affiliation for a word. Further research is needed to study the effects of adding features to the classifier training.

Regardless, an interesting finding is that part of speech affiliation is distributed among many components of the word embeddings, not concentrated in one or two specific features. Thus, the strongly expressed component 94 in the average vector of conjunctions (Figures 1 and 2) seems to be a solitary case.

## 6 Conclusion

We showed that semantic features derived in the process of training distributional vector models, can be employed both in supporting linguistic hypotheses about part of speech class changes and in detecting and fixing possible annotation errors in corpora. Word embeddings contain rather robust information about the PoS class of the corresponding words. Moreover, this knowledge seems to be distributed among several components (at least a hundred in our case of 300-dimensional model).

Distributional models trained in a non-deterministic and stochastic way on large amounts of word contexts learn knowledge about part of speech clusters. Arguably, they are good at this precisely because part of speech boundaries are not strict, and even sometimes considered to be a non-categorical linguistic phenomenon (Manning, 2015).

The reported experiment form part of ongoing research, and we plan to extend it, particularly conducting similar experiments with other languages typologically different from English. We also plan to continue studying the issue of correspondence between particular embedding components and part of speech affiliation. Another direction of future work is finding out how different hyperparameters for training distributional models (including training corpus pre-processing) influence their performance in PoS discrimination.

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
