# Peer review of "Redefining part-of-speech classes with distributional semantic models"

_CoNLL 2016 — decision unknown_

[Official Review · Reviewer 1 · rating 2 · confidence 4]
soundness 4 · originality 2 · clarity 4 · impact 2 · substance 1 · appropriateness 3 · meaningful comparison 3 · replicability 5 · presentation format Poster

The aim of this paper is to show that distributional information stored in word
vector models contain information about POS labels. They use a version of the
BNC annotated with UD POS and in which words have been replaced by lemmas. They
train word embeddings on this corpus, then use the resulting vectors to train a
logistic classifier to predict the word POS. Evaluations are performed on the
same corpus (using cross-validation) as well as on other corpora. Results are
clearly presented and discussed and analyzed at length.

The paper is clear and well-written. The main issue with this paper is that it
does not contain anything new in terms of NLP or ML. It describe a set of
straightforward experiments without any new NLP or ML ideas or methods. Results
are interesting indeed, in so far that they provide an empirical grounding to
the notion of POS. In that regard, it is certainly worth being published in a
(quantitative/emprirical) linguistic venue.

On another note, the literature on POS tagging and POS induction using word
embeddings should be cited more extensively (cf. for instance Lin, Ammar, Duer
and Levin 2015; Ling et al. 2015 [EMNLP]; Plank, SÃ¸gaard and Goldberg
2016...).

[Official Review · Reviewer 2 · rating 2 · confidence 5]
soundness 4 · originality 2 · clarity 4 · impact 2 · substance 4 · appropriateness 5 · meaningful comparison 1 · replicability 4 · presentation format Poster

## General comments:
This paper presents an exploration of the connection between part-of-speech
tags and word embeddings. Specifically the authors use word embeddings to draw
some interesting (if not somewhat straightforward) conclusions about the
consistency of PoS tags and the clear connection of word vector representations
to PoS. The detailed error analysis (outliers of classification) is definitely
a strong point of this paper.

However, the paper seems to have missing one critical main point: the reason
that corpora such as the BNC were PoS tagged in the first place. Unlike a
purely linguistic exploration of morphosyntactic categories (which are
underlined by a semantic prototype theory - e.g. see Croft, 1991), these
corpora were created and tagged to facilitate further NLP tasks, mostly
parsing. The whole discussion could then be reframed as whether the
distinctions made by the distributional vectors are more beneficial to parsing
as compared to the original tags (or UPOS for that matter). 

Also, this paper is missing a lot of related work in the context of
distributional PoS induction. I recommend starting with the review
Christodoulopoulos et al. 2010 and adding some more recent non-DNN work
including Blunsom and Cohn (2011), Yatbaz et al. (2012), etc. In light of this
body of work, the results of section 5 are barely novel (there are systems with
more restrictions in terms of their external knowledge that achieve comparable
results).

## Specific issues
In the abstract one of the contributed results is that "distributional vectors
do contain information about PoS affiliation". Unless I'm misunderstanding the
sentence, this is hardly a new result, especially for English: every
distributionally-based PoS induction system in the past 15 years that presents
"many-to-one" or "cluster purity" numbers shows the same result.

The assertion in lines 79-80 ("relations between... vectors... are mostly
semantic") is not correct: the <MIKOLOV or COLOBERT> paper (and subsequent
work) shows that there is a lot of syntactic information in these vectors. Also
see previous comment about cluster purity scores. In fact you revert that
statement in the beginning of section 2 (lines 107-108).

Why move to UPOS? Surely the fine-grained distinctions of the original tagset
are more interesting.

I do not understand footnote 3. Were these failed attempts performed by you or
other works? Under what criteria did they fail? What about Brown cluster
vectors? They almost perfectly align with UPOS tags.

Is the observation that "proper nouns are not much similar to common nouns"
(lines 331-332) that interesting? Doesn't the existence of "the" (the most
frequent function word) almost singlehandedly explain this difference?

While I understand the practical reasons for analysing the most frequent
word/tag pairs, it would be interesting to see what happens in the tail, both
in terms of the vectors and also for the types of errors the classifier makes.
You could then try to imagine alternatives to pure distributional (and
morphological - since you're lemmatizing) features that would allow better
generalizations of the PoS tags to these low-frequency words.

## Minor issues
Change the sentential references to \newcite{}: e.g. "Mikolov et al. (2013b)
showed"